# Treatment Strategies for Non-Small Cell Lung Cancer Harboring Common and Uncommon EGFR Mutations: Drug Sensitivity Based on Exon Classification, and Structure-Function Analysis

**DOI:** 10.3390/cancers14102519

**Published:** 2022-05-20

**Authors:** Rui Kitadai, Yusuke Okuma

**Affiliations:** 1Department of Medical Oncology, National Cancer Center Hospital, Tokyo 104-0045, Japan; rkitadai@ncc.go.jp; 2Department of Thoracic Oncology, National Cancer Center Hospital, Tokyo 104-0045, Japan

**Keywords:** common *EGFR* mutation, uncommon *EGFR* mutation, non-small cell lung cancer, EGFR-TKI, structural feature

## Abstract

**Simple Summary:**

The advent of epidermal growth factor receptor (*EGFR*) tyrosine kinase inhibitors (EGFR-TKIs) has led to a dramatic improvement in the prognosis of patients having advanced *EGFR*-mutant non-small cell lung cancer (NSCLC). NSCLCs harboring “common” *EGFR* mutations, including exon 19 deletions and exon 21 L858R mutation substitutions, are sensitive to EGFR-TKIs. However, NSCLCs harboring “uncommon” *EGFR* mutations have poor sensitivity to EGFR-TKIs, and patients harboring uncommon mutations often experience poor outcomes. Here, we review the current EGFR-TKI therapy and the development of treatment strategies, including combined treatment and the exploration of new drugs. In addition, we discuss EGFR-TKI sensitivity based on structure-function analysis.

**Abstract:**

The identification of epidermal growth factor receptor (*EGFR*) mutations and development of *EGFR* tyrosine kinase inhibitors (EGFR-TKIs) have dramatically improved the prognosis of advanced EGFR-mutated non-small cell lung cancer (NSCLC), setting a landmark in precision oncology. Exon 19 deletions and exon 21 L858R substitutions, which comprise the majority of common *EGFR* mutations, are predictors of good sensitivity to EGFR-TKIs. However, not all cancers harboring *EGFR* mutations are sensitive to EGFR-TKIs. Most patients harboring uncommon *EGFR* mutations demonstrate a poorer clinical response than those harboring common *EGFR* mutations. For example, cancers harboring exon 20 insertions, which represent approximately 4–12% of *EGFR* mutations, are generally insensitive to first- and second-generation EGFR-TKIs. Although understanding the biology of uncommon *EGFR* mutations is essential for developing treatment strategies, there is little clinical data because of their rarity. Moreover, clarifying the acquired resistance of *EGFR*-mutated NSCLC may lead to more precise treatments. Sequencing and structure-based analyses of EGFRmutated NSCLC have revealed resistance mechanisms and drug sensitivity. In this review, we discuss the strategies in development for treating NSCLC harboring common and uncommon *EGFR* mutations. We will also focus on EGFR-TKI sensitivity in patients harboring *EGFR* mutations based on the structural features.

## 1. Introduction

Activating epidermal growth factor receptor (*EGFR*) mutations are observed in approximately 10–15% of Caucasian patients and 30–35% of Asian patients with non-small cell lung cancer (NSCLC) [1]. Among these mutations, 40–50% are deletions in exon 19, and 30–40% are L858R substitutions in exon 21. *EGFR* tyrosine kinase inhibitors (TKIs) have been a standard of care as the first-line pharmacotherapy for advanced NSCLC. Many countries have approved three generations of EGFR-TKIs. First-generation EGFR-TKIs include gefitinib and erlotinib, second-generation EGFR-TKIs include afatinib and dacomitinib, and third-generation EGFR-TKIs include osimertinib. Several clinical trials have shown that gefitinib, erlotinib, and afatinib showed superiority in efficacy to platinum-based chemotherapy as a standard of care for treatment-naïve NSCLC patients harboring *EGFR* mutations [2,3,4,5]. However, most patients develop resistance and progress after a median of 9–13 months of EGFR-TKI monotherapy. Several acquired resistance mechanisms have been investigated: acquired mutations such as T790M and T790S mutations, alternative signaling activation (Met, HGF, AXL, Hh, IGF-1R), aberrant downstream pathways, impairment of the EGFR-TKI-mediated apoptosis pathway, and histological transformation [6]. Osimertinib was developed to selectively inhibit sensitizing and target T790M mutations in NSCLC patients. Compared with first- or second-generation EGFR-TKIs, osimertinib has remarkable efficacy as a first-line treatment with a median progression-free survival (PFS) and overall survival (OS) of 18.9 and 38.6 months, respectively. In standard-of-care first-generation EGFR-TKI, the median PFS and OS are 10.2 and 31.8 months, respectively [7,8]. However, patients have shown disease progression after approximately 18.9 months of first-line treatment with osimertinib. Therefore, additional treatment strategies are still required.

In addition to exon 19 deletions and exon 21 L858R substitutions, uncommon *EGFR* mutations can be found in 10–15% of patients harboring EGFR-mutated NSCLC [9]. Uncommon *EGFR* mutations show various efficacy to EGFR-TKI depending on the molecular alterations, which are not yet completely understood as they have often been excluded from clinical trials. Exon 20 insertions are the third most common subtype of *EGFR* mutations. They are exclusive to other driver mutations and are usually not sensitive to approved EGFR-TKIs. Many clinical trials targeting patients with exon 20 insertions are ongoing, which may provide effective targeted agents.

Due to the varying sensitivities to EGFR-TKIs in NSCLC patients with *EGFR* mutations, an attempt has been made to establish a classification system to accurately predict drug sensitivity. Structure-function analysis has been performed, and recently, classifications based on structural and functional changes have been reported. In this article, we review recent studies and discuss the development of systematic treatments for NSCLC harboring common and uncommon *EGFR* mutations.

## 2. Clinical Features and Molecular Characteristics of EGFR-Mutant NSCLC

*EGFR* mutations are known to occur most frequently in adenocarcinoma in females, non-smokers, and the Asian population [10]. Approximately 47% of patients in Pacific Asian countries harbor *EGFR* mutations [11]. According to a meta-analysis conducted to characterize the patterns of mutation incidence in NSCLC, the frequency of *EGFR* mutations was 47.9% in adenocarcinoma (ADC) and 4.6% in squamous cell carcinoma (SCC) in Asian populations. The frequencies of *EGFR* mutations in Western populations were 19.2% in ADC and 3.3% in SCC [6].

*EGFR* is a receptor tyrosine kinase that is a member of the *ErbB* family. The *ErbB* family comprises *ErbB1* (*EGFR* or *HER1*), *ErbB2* (*HER2* or *Neu*), *ErbB3* (*HER3*), and *ErbB4* (*HER4*). *EGFR* consists of an extracellular ligand-binding domain, transmembrane domain, juxtamembrane domain, kinase domain, and carboxy-terminal (C-terminal) signaling domain [12]. The ligand binding to the extracellular domain triggers dimerization, autophosphorylation, and the binding of adaptor proteins. This enables the ligand to interact with signaling components in downstream signaling pathways, such as the RAS-MAPK, PI3K-AKT, and STAT pathways, to enhance cell proliferation and inhibit apoptosis, angiogenesis, and migration [13,14]. *EGFR* mutations are mostly present in the first four exons, which are exons 18–21, of the gene encoding the tyrosine kinase domain (Figure 1). Approximately 90% of the *EGFR* mutations are either exon 19 deletions, deletions encompassing amino acids from codons 746 through 750, or L858R, and missense mutations resulting in leucine to arginine at codon 858 [10]. There are over 20 variants of deletions, such as larger deletions, deletions plus point mutations, and deletions plus insertions. Approximately 3% of these mutations are G719X, which occurs at codon 719, resulting in the substitution of glycine with cysteine, alanine, or serine. Additionally, the aforementioned mutations, S768I, exon 20 insertions, and L861Q, often occur. The first-, second-, and third-generation EGFR-TKIs used in clinical practice are competitive inhibitors of ATP. They show efficacy by binding to the ATP-binding pocket in the kinase domain. Their binding to mutant *EGFR* is stronger than that to the wild-type receptor. Drug resistance may quickly arise, however, due to the acquired mutation of T790M, which occurs in approximately half of the patients treated with EGFR-TKIs [15,16]. Since T790 is located at the base of the ATP binding site and close to the adenosine moiety of ATP, the T790M mutation increases the affinity of the mutant receptor for ATP [17]. Osimertinib, an irreversible EGFR-TKI, was designed to selectively inhibit sensitizing and T790M mutations.

## 3. Treatment for Common *EGFR* Mutations

### 3.1. EGFR-TKI

Most clinical trials have been designed for NSCLC patients harboring common *EGFR* mutations, including exon 19 deletions and exon 21 L858R point mutations. Phase 3 trials of first-generation (gefitinib and erlotinib) or second-generation EGFR-TKIs (afatinib) compared with platinum doublets have shown superior efficacy, improved PFS, and have become the standard first-line pharmacotherapy for NSCLC harboring common *EGFR* mutations [2,4,5,20]. Dacomitinib, a second-generation EGFR-TKI, was compared with gefitinib in the phase 3 AHCHER-1050 trial. Although dacomitinib showed clinical benefits compared to gefitinib for previously untreated EGFR-mutated NSCLC patients without CNS metastasis, treatment-related serious adverse events were observed in 9% of patients, and dose reduction was reported in 66% [21]. The third-generation EGFR-TKI, osimertinib, was designed to selectively inhibit the sensitizing and T790M mutant forms of *EGFR* tyrosine kinase, which is one of the acquired mutations to first-generation and second-generation EGFR-TKIs. An open-label phase 3 study was conducted to compare osimertinib with platinum-based chemotherapy in advanced T790M-positive NSCLC patients who exhibited disease progression after first-line EGFR-TKI therapy. Patients who received osimertinib had superior PFS to patients treated with chemotherapy (10.1 vs. 4.4 months) (AURA 3 trial) [22]. In addition, osimertinib demonstrated superior efficacy in the central nervous system (CNS) to platinum-based chemotherapy. However, osimertinib is not approved for T790M-negative patients with disease progression after first- or second-generation EGFR-TKI administration. A phase 2 study (WJOG12819L) was conducted to assess the efficacy of osimertinib in patients with T790M-negative or unknown disease who developed isolated CNS progression or systemic disease progression during first- or second-generation EGFR-TKIs. In the TREM study, a phase 2 study for previously treated NSCLC harboring T790M-negative *EGFR* mutations treated with progression on at least one previous EGFR-TKI, the overall response rate was 28% (95% CI, 15–41%), median PFS was 5.1 months, and median OS was 13.4 months, respectively [23]. The efficacy of osimertinib in the first-line treatment of NSCLC patients with the *EGFR* mutation was evaluated in a randomized, double-blind, phase 3 FLAURA study [7]. Patients were assigned either osimertinib or comparator first-generation EGFR-TKIs (erlotinib or gefitinib) and were stratified by *EGFR* mutations (exon 19 deletion or L858R point mutation) and race. Compared with first-generation TKI, osimertinib presented superior efficacy with a median PFS and OS of 18.9 and 38.6 months, respectively, and less grade 3 or higher adverse events [8]. Based on these results, osimertinib may be considered as a standard first-line pharmacotherapy for common *EGFR* mutation-positive NSCLC.

Both exon 19 deletions and L858R point mutations show high sensitivity to EGFR-TKIs; however, it has been reported that patients with these two mutations have different clinical outcomes. A meta-analysis of 12 clinical trials was conducted to calculate the efficacy of EGFR-TKIs between the two mutations. Patients with exon 19 deletions had a significantly longer PFS (hazard ratio (HR) = 0.69; 95% CI (0.57, 0.82); *p* < 0.001) and OS (HR = 0.61; 95% CI (0.43, 0.86); *p* = 0.005). Their overall response rates (ORR) were higher (odds ratio = 2.14; 95% CI (1.63, 2.81); *p* < 0.001) compared to those with exon 21 L858R mutations [24]. In the FLAURA trial, first-line osimertinib treatment was associated with a longer PFS in patients with exon 19 deletions (21.4 months) than in those with the L858R mutation (14.4 months) [7]. 

CNS metastases are observed in approximately 31% of NSCLC patients harboring *EGFR* mutations [25]. First- and second-generation EGFR-TKIs have limited efficacy against CNS metastases [25,26,27,28]. Preclinical studies have demonstrated that osimertinib has higher activity than other EGFR-TKIs in EGFR-mutant NSCLC brain metastasis models [29]. Clinical trials of osimertinib have also shown its efficacy in CNS metastases. In the AURA3 trial, the CNS ORR in patients with one or more measurable CNS lesions was 70% with osimertinib. The median CNS PFS in patients with measurable and/or non-measurable CNS lesions was 11.7 months [30]. In the FLAURA trial, the CNS ORR for patients receiving osimertinib was 91%. The median CNS PFS was not reached, whereas the median CNS PFS of patients receiving first-generation TKI was 13.9 months [31]. Recently, clinical trials, including the TORG1938/EPONA study [jRCTs071200029] and the COMPEL study [32], have been ongoing for EGFR-mutant NSCLC patients harboring CNS metastasis who developed systemic disease progression but stable CNS metastases during treatment with osimertinib. Patients will receive either platinum-based chemotherapy plus osimertinib or platinum-based chemotherapy alone to evaluate the efficacy for CNS metastases.

### 3.2. EGFR-TKI Combined with Chemotherapy

The majority of EGFR-mutant NSCLC develop resistance and progress after 9–13 months of EGFR-TKI monotherapy. Platinum-doublet chemotherapy is the recommended treatment option after targeted treatment. Therefore, additional treatment options are needed to enhance the long-term efficacy of EGFR-TKIs. Recently, the efficacy of combination therapy with EGFR-TKIs has been reported. A clinical trial of gefitinib combined with carboplatin plus pemetrexed was conducted for chemotherapy-naive advanced or relapsed non-squamous NSCLC harboring *EGFR* mutations (NEJ009) in Japan [33]. Median PFS was significantly longer in the combination arm than in the gefitinib arm (20.9 vs. 11.2 months, respectively; HR = 0.49; 95% CI (0.39, 0.62); *p* < 0.001), however, PFS2 did not show superiority. The combination arm also had a better ORR than the gefitinib arm (84% vs. 67%, respectively; *p* < 0.001), and the exploratory study showed that OS was 50.9 months in the combination arm vs. 38.8 months in the gefitinib arm, which indicated that it might be an effective new treatment option. Another phase 3 trial assessing the efficacy of gefitinib plus carboplatin and pemetrexed in India also showed better results in the combined therapy arm. The ORR of the combined therapy arm vs. the gefitinib arm was 75% vs. 63% (*p* = 0.01), the PFS was 16 vs. 8 months (HR = 0.51), and the OS results were not reached vs. 17 months (HR = 0.45) [34]. One ongoing clinical trial is a phase 3 study for patients with advanced non-squamous NSCLC harboring EGFR-activating mutations. The study compares EGFR-TKI alone (gefitinib or osimertinib) and EGFR-TKI with three intercalated cycles of cisplatin + pemetrexed (JCOG1404/WJOG8214L). This study was planned on the basis of the hypothesis that the administration of platinum-doublet chemotherapy with EGFR-TKIs prevents the emergence of acquired resistance to EGFR-TKIs, and may prolong patients’ survival [35]. Another ongoing phase 3 randomized trial evaluated the efficacy and safety of osimertinib with platinum-pemetrexed compared to osimertinib monotherapy in first-line treatment for *EGFR* mutation-positive advanced or metastatic NSCLC patients (FLAURA2 study/NCT04035486). A safety analysis of the run-in phase has been reported [36]. Adverse events (AEs) were reported in 90% of patients, and the most common AEs were constipation and nausea. Twenty percent of the patients reported serious AEs. Although one patient discontinued treatment due to pneumonitis, a manageable safety profile was observed as a whole. Moreover, second-line generation combination therapy research for patients who acquired resistance to osimertinib is ongoing, such as the phase 2 trial of afatinib, carboplatin, and pemetrexed (NEJ025B), and the phase 1 trial of dacomitinib plus osimertinib (NCT03755102). 

Recently, the results of EGFR-TKI combined with anti-angiogenic drugs have been reported in several clinical trials. Erlotinib plus bevacizumab showed the efficacy and manageable toxicity in treatment-naïve patients with NSCLC compared with erlotinib monotherapy in the JO25567 phase 2 study [37]. Subsequently, a phase 3 study, NEJ026, was performed including patients with CNS metastases [38]. At the time of interim analysis, patients in the erlotinib plus bevacizumab arm showed the median PFS of 16.9 months (95% CI [14.2, 21.0]), in comparison with the PFS of patients in the erlotinib arm of 13.3 months (95% CI (11.1, 15.3)) (HR = 0.605, 95% CI (0.417, 0.877); *p* = 0.016). In the subgroup analysis, patients with the L858R mutation, which showed less efficacy than exon 19 deletions in EGFR-TKI monotherapy across many trials, showed a longer median PFS than those with exon 19 deletions. Moreover, the subgroup analysis suggested that adding bevacizumab may prolong the median PFS for patients with malignant pleural effusion or pleural metastases. However, follow-up OS analysis showed no significant difference in the median OS between the two arms [39]. This may be due to long post-progression survival, which reduces the impact of first-line therapy. A randomized phase 3 trial of ramucirumab plus erlotinib in patients with treatment-naive, EGFR-mutated, advanced NSCLC was also reported [15]. Although this trial excluded patients with CNS metastases, PFS was significantly longer in the ramucirumab plus erlotinib arm (19.4 months, 95% CI (15.4, 21.6)) compared with the placebo plus erlotinib arm (12.4 months, 95% CI (11.0, 13.5)) (HR = 0.59, 95% CI (0.46, 0.76); *p* < 0.0001). Regarding osimertinib, a phase 2 randomized trial for patients harboring a “common” *EGFR* mutation has been reported. However, osimertinib plus bevacizumab did not show improved PFS compared to osimertinib alone (UMIN000030206).

### 3.3. Immune Checkpoint Inhibitors in EGFR-Mutated NSCLC

The role of immune checkpoint inhibitor (ICI) monotherapy, programmed death-1 (PD-1), and programmed death ligand 1 (PD-L1) inhibitors in NSCLC patients harboring *EGFR* mutations is limited. In a phase 2 trial of pembrolizumab for TKI-naive and PD-L1-positive NSCLC patients harboring *EGFR* mutations, the response rate was 0%, which led to early discontinuation of the trial [40]. A subgroup analysis of another phase 2 study was conducted on EGFR-mutant patients treated with atezolizumab in different lines of therapy (BIRCH trial) [41]. With first-line therapy, the response was 19% among the 13 EGFR-mutant patients, which was similar to that of *EGFR* wild-type patients. Among the 32 remaining EGFR-mutant patients administered second-line or above treatment, there was little to no response. A phase 2 study evaluating the efficacy of durvalumab for pretreated NSCLC patients was also conducted (ATLANTIC trial). Among patients harboring *EGFR* mutations, outcomes were inferior to those of patients with wild-type *EGFR* regardless of PD-L1 expression [42]. A meta-analysis assessing the efficacy of PD-1/PD-L1 inhibitors in EGFR-mutant NSCLC patients as second-line therapy, including four trials (CheckMate 057, KEYNOTE-010, POPLAR, and OAK study), concluded that patients with *EGFR* mutations did not show OS benefit from ICIs over docetaxel [43].

A randomized phase 3 trial was conducted for PD-L1 inhibitor plus chemotherapy. Chemotherapy-naive patients with advanced NSCLC were treated with either atezolizumab, bevacizumab, carboplatin, and paclitaxel (ABCP); atezolizumab, carboplatin, and paclitaxel (ACP); or bevacizumab, carboplatin, and paclitaxel (BCP) (IMpower 150 trial) [44]. Efficacy was assessed in the key subgroups, including patients harboring *EGFR* mutations (both sensitizing and non-sensitizing) who had previously been treated with one or more EGFR-TKIs. The analysis showed an improved OS with sensitizing EGFR-mutated patients treated with ABCP compared to those treated with BCP [45]. These data indicate that ABCP may be a therapeutic option for EGFR-mutated NSCLCs after the failure of first-line EGFR-TKI therapy in patients who do not acquire the secondary T790M mutation. Moreover, a phase 2 study of toripalimab plus chemotherapy in EGFR-mutant advanced NSCLC patients who underwent failed prior EGFR-TKI therapy showed a high ORR of 50% and median PFS of 7 months [46]. The KEYNOTE-789 [47], CheckMate 722 [48], and WJOG8515L [49] trials are currently enrolling patients with EGFR-mutant advanced NSCLC whose diseases progressed after prior EGFR-TKI treatments. The patients are assigned to receive either chemotherapy alone or chemotherapy combined with PD-1/PD-L1 inhibitor.

Clinical data on the combination of PD-1 and CTLA-4 inhibitors for EGFR-mutant NSCLC patients are currently insufficient. In the CheckMate 012 trial, ORR was 50% in patients with *EGFR* mutations who received nivolumab combined with ipilimumab, however the data were from eight patients [50]. Cohorts D and H of the KEYNOTE-021 trial reported that ORR was only 10% in TKI-pretreated NSCLC patients harboring *EGFR* mutation who underwent combined treatment with pembrolizumab and ipilimumab [51].

## 4. Treatment for Uncommon *EGFR* Mutations (Other Than Exon 20 Insertions)

### 4.1. Treatment for Major Uncommon Mutations in NSCLC

*EGFR* exon 20 insertions comprise approximately half of the uncommon *EGFR* mutations. In addition to exon 20 insertions, the major and most prevalent uncommon *EGFR* mutations are G719X, S768I, and L861Q, which are known as “major uncommon mutations” [18]. These mutations account for approximately 3.1%, 1.1%, and 0.9% of all *EGFR* mutations in NSCLC, respectively [52]. Up to 25% of uncommon positive *EGFR* mutations coexist with other *EGFR* mutations, termed “compound mutations” [53]. The data of in vitro drug sensitivities are summarized in Table 1 [52,54].

Clinical data for uncommon *EGFR* mutations are limited as these patients have generally been excluded from randomized trials. A combined analysis of LUX-Lung 2, LUX-Lung 3, and LUX-Lung 6, in which patients harboring uncommon mutations were treated with afatinib, showed that exons 18, 16, and 8 harbored G719X, L861Q, and S768I mutations, respectively. The corresponding response rates were 78%, 56%, and 100%. The median PFS was 13.8, 8.2, and 14.7 months, respectively [55]. Based on these findings, afatinib was approved for NSCLC patients harboring sensitizing *EGFR* mutations by the U.S. Food and Drug Administration (FDA). Recently, a pooled analysis was conducted to assess the activity of afatinib [18]. Among the 315 EGFR-TKI-naive patients treated with afatinib, 62, 55, and 10 had G719X, L861Q, and S768I mutations, and the ORR was 63%, 60%, and 63%, respectively. The median time-to-treatment failure (TTF) was 14.7, 10.0, and 15.6 months, respectively. On the other hand, gefitinib has shown moderate sensitivity in patients harboring major uncommon mutations compared to afatinib. A sub-analysis of a phase 3 trial of gefitinib (NEJ-002) suggested that the G719X and L861Q mutations were less sensitive to gefitinib, with a response rate of 20% [56]. According to the COSMIC database, the ORR of patients with G719X, S768I, and L861Q treated with gefitinib was 32%, 42%, and 39%, respectively [52]. In a retrospective study in China, the outcomes of patients with G719X, L861Q, and S768I mutations treated with erlotinib/gefitinib had an ORR of 37%, 40%, and 33%, respectively [57].

A phase 2 study of osimertinib was conducted in 37 patients with uncommon *EGFR* mutations, excluding the exon 20 insertions [58]: 53% had the G719X mutation, 25% had the L861Q mutation, 22% had the S768I mutation, and 11% had other mutations. Overall, an ORR of 50% was observed. The median PFS was 8.2 months, and the median OS was not reached for all 36 evaluated patients. The response rates in patients with the G719X, L861Q, and S768I mutations were 53%, 78%, and 38%, respectively. The median PFS was 8.2, 15.2, and 12.3 months, respectively. Thirty-nine percent of the enrolled patients had previously been treated with chemotherapy. The safety profile of osimertinib in this study was acceptable and mostly confined to grade 1–2 AEs. Additionally, a retrospective study reported the efficacy of osimertinib in 51 patients harboring uncommon mutations [59]. Twenty patients were treatment-naive. In this analysis, the median times of treatment for patients harboring G719X and L861Q mutations were 5.8 and 19.3 months, respectively. The AURA trial examined seven EGFR-TKI-pretreated patients harboring G719X mutations treated with osimertinib. Three patients had an additional T790M mutation. Among the three patients harboring G719X and T790M mutations, two exhibited a partial response, and one had stable disease. Only one of the four patients without the T790M mutation showed a response [60]. Recently, a phase 2 study of osimertinib for uncommon or compound EGFR-mutated previously untreated NSCLC has been ongoing (UNICORN study) [61]. The efficacy of ICIs has been evaluated in EGFR-mutated NSCLCs, as mentioned in Section 3.3; however, there is a lack of data regarding patients who harbor uncommon mutations.

### 4.2. Treatment for Other Uncommon EGFR-Mutated NSCLC

Although the data are limited, the efficacy of EGFR-TKIs has been reported for uncommon mutations, apart from previously reported uncommon mutations, exon 20 insertions, and T790M mutations. The exon 18 mutations have modest sensitivity to first-generation EGFR-TKI with a response rate of approximately 50% [52,62]. Three out of eight patients with E709X responded to afatinib, and two patients with Del18 mutations responded according to the database [18]. Exon 19 mutations include rare substitution mutations including L747P/S, insertions such as I744_K745KIPVAI, and insertion-deletions including L747_A750 > *p*. It is reported that L747P/S mutations are resistant to first-generation EGFR-TKIs but may be sensitive to afatinib [63]. Preclinical data have shown greater sensitivity to afatinib and osimertinib than first-generation TKIs for exon 20 (apart from S768I and T790M mutations) [64]. For exon 21 mutations, afatinib and osimertinib were more effective than first-generation TKIs in preclinical data [64].

### 4.3. Compound EGFR Mutations in NSCLC

Recent data have shown that up to 25% of patients with *EGFR* mutation-positive NSCLC harbor compound mutations [53]. Since this subgroup includes patients harboring co-occurring *EGFR* mutations, coexisting common and uncommon *EGFR* mutations, and uncommon mutations only, the sensitivity varies. In general, if both *EGFR* mutations are known to be sensitive to EGFR-TKI, sensitivity is generally similar to that of solitary uncommon *EGFR* mutations. However, if sensitive and resistant mutations coexist, the sensitivity will be reduced [65]. 

Compound *EGFR* mutations were identified in 40 patients in the afatinib uncommon *EGFR* mutation database [18]. Among these patients, 26 out of 40 had at least 1 major uncommon *EGFR* mutation. Patients were treated with afatinib as a first-line treatment, and in assessable patients, the ORR was 77%. The median duration of response (DoR) was 16.6 months. In patients harboring a major uncommon *EGFR* mutation, the ORR was 78%, and the median DoR was 17.1 months. With osimertinib, four patients had compound *EGFR* mutations (two harboring G719X þ L861Q and two with S768I þ G719X) in the KCSG-LU15-09 trial, and three out of four patients showed a response [58]. Some real-world data have suggested that patients harboring compound *EGFR* mutations exhibit a fair response to first-generation EGFR-TKIs. 

A cohort of 46 patients harboring compound *EGFR* mutations treated with gefitinib, erlotinib, or afatinib exhibited a median PFS and OS of 12.3 and 31.0 months, respectively [66]. However, if the compound *EGFR* mutation included a common *EGFR* mutation, the outcome would be better. Moreover, in a cohort study of 11 patients, 9 of whom had compound mutations comprising 2 uncommon *EGFR* mutations, the ORR was 82% and the median PFS was 5.1 months. In this study, eight patients had a major uncommon *EGFR* mutation: six had the G719X mutation and other *EGFR* mutations, and two had the S768I mutation and other *EGFR* mutations [67]. However, some studies have indicated that although Del19 or L858R mutations exist, the presence of an uncommon *EGFR* mutation may reduce the responsiveness to first-generation EGFR-TKIs. In a retrospective study, the response rate to gefitinib of 11 patients harboring compound *EGFR* mutations was 18% [68]. Another study of 20 patients showed an ORR and median PFS of 60% and 5.3 months, respectively [69]. First-generation EGFR-TKIs may be a treatment option; however, according to these results, afatinib and osimertinib need to be considered for patients harboring compound *EGFR* mutations depending on their specific cases. These treatments, therefore, require further study. 

### 4.4. Treatment for De Novo T790M Mutations in NSCLC

The de novo T790M previously untreated *EGFR* mutation has been detected using highly sensitive methods and is usually associated with shorter PFS [70,71,72]. The AURA 3 trial showed superior PFS in T790M-positive NSCLC compared to patients treated with chemotherapy, as previously mentioned in Section 3.1 [22]. Recently, two randomized phase 2 studies evaluating the efficacy of adding bevacizumab to osimertinib in NSCLC patients with acquired *EGFR* T790M mutation have been reported. However, in both studies, the combination arm did not show a prolonged PFS [19,73].

## 5. Development of Treatment in *EGFR* Exon 20 Insertions

*EGFR* exon 20 insertions represent approximately 4–12% of *EGFR* mutations in patients with NSCLC [9,74,75,76]. Among these mutations, V769_D770insASV, D770_N771insSVD, and A763_Y764insFQEA account for 20%, 19%, and 7% of the mutations, respectively [52]. In general, exon 20 insertions are not sensitive to first- and second-generation EGFR-TKIs [9]. The sensitivity of exon 20 insertions to first-generation EGFR-TKI and afatinib are known to be poor, with an ORR of 17% and 10%, respectively [76,77,78,79,80,81,82]. On the other hand, A763_Y764insFQEA is reported to have sensitivity to gefitinib, erlotinib, afatinib, and osimertinib [83]. Osimertinib showed a PFS of 4.2 months for A763_Y764insFQEA [84]; however, its efficacy is limited, and drug development focusing on exon 20 insertion is needed. 

Several EGFR-TKIs have been investigated, including mobocertinib (TAK-788), poziotinib, TAS6417, and BDTX-189. Mobocertinib is an oral irreversible *EGFR* and *HER2* TKI designed to selectively target in-frame *EGFR* exon 20 insertion. A phase 1/2 study of 28 patients with previously treated *EGFR* exon 20 insertion-positive NSCLC showed investigator-assessed confirmed ORR of 43% and median PFS of 7.3 months [85]. Recently, the results of 114 platinum-pretreated patients with exon 20 insertion-positive NSCLC in the dose-escalation, expansion, and EXCLAIM cohorts were reported [86]. The confirmed ORR was 23% by a blinded independent review and 32% by an investigator review, with a median PFS of 7.3 months and OS of 24.0 months. The most common AEs were diarrhea and rash. Diarrhea was the only grade 3 or 4 treatment-related AE reported in more than 10% of patients. Based on these efficacies, mobocertinib was approved by the FDA as the first oral therapy for NSCLC patients with *EGFR* exon 20 insertion. For treatment-naive patients, a phase 3 EXCLAIM-2 study is ongoing to evaluate the efficacy of mobocertinib vs. platinum-doublet chemotherapy (NCT04129502). Poziotinib is also an *EGFR* and *HER2* TKI whose structure is similar to that of the second-generation EGFR-TKI afatinib [87]. A phase 2 study (ZENITH20-1) was conducted in patients with either *EGFR* or *HER2* exon 20 insertions with at least one prior line of therapy [88]. Among the 115 patients, the ORR was 14.8%, which did not meet the primary endpoint. The disease control rate (DCR) was 68.7% and the median PFS was 4.2 months. High rates of grade ≥ 3 AEs of rash (28%) and diarrhea (26%) were observed in 65% of patients, requiring dose reduction from the starting dose of 16 mg daily. Ongoing cohorts of the ZENITH20 trial are exploring alternative dosing strategies. TAS6417 is an oral EGFR-TKI with broad activity against clinically relevant *EGFR* mutations, including ins20. The interim results of a phase 1/2a trial (NCT04036682) have also been reported. Among the 25 response-evaluable patients, 40% exhibited a partial response, 56% had stable disease, and 4% had progressive disease as the best response [89]. BDTX-189 has an inhibitory effect on *EGFR* and *HER2* ins20 mutations. A phase 1/2 clinical trial (NCT04209465) for patients with advanced solid tumors harboring *EGFR* or *HER2* ins20 mutations is currently in progress.

Amivantamab is an IgG1-based bispecific antibody targeting both *EGFR* and the mesenchymal–epithelial transition factor (MET) [90]. Its antitumor activity functions by disrupting *EGFR* and MET signaling functions by blocking ligand binding, producing lysosomal degradation of *EGFR* and MET receptors, and inducing Fc-mediated antibody-dependent cellular cytotoxicity (ADCC) and antibody-dependent cell-mediated phagocytosis (ADCP) [90,91]. CHRYSALIS (NCT02609776) is a phase 1 study for patients with EGFR-mutant advanced NSCLC harboring a variety of different *EGFR* mutations [92]. The safety profile was reported for 50 patients harboring exon 20 insertions who received 1050 mg of amivantamab. The most common AEs reported were rash (72%), infusion-related reactions (60%), and paronychia (34%). Grade ≥ 3 treatment-related AEs were reported at 6%. No grade ≥ 3 rashes were reported. The ORRs were between 36% and 41% in the 29 patients previously treated with platinum-based chemotherapy. Updated results for the post-platinum *EGFR* exon 20 insertion NSCLC population of CHRYSALIS treated with 1050 mg of amivantamab were presented [93]. Among 114 patients in the safety population, the most common AE was rash in 86% of patients, with only 4% exhibiting a grade 3 of rash, infusion reactions (66%), and paronychia (45%). Treatment-related dose reductions and discontinuations were reported in 13% and 4% of patients, respectively. Among the 81 patients in the efficacy population, the ORR was 40% with a median PFS of 8.3 months. Amivantamab became the first targeted therapy of NSCLC patients with *EGFR* exon 20 insertions, and was granted FDA breakthrough therapy designation and became the first therapy to be granted FDA approval in May 2021. A phase 3 trial is ongoing for patients with exon 20 insertions to evaluate the efficacy of amivantamab plus chemotherapy vs. chemotherapy alone (PAPILLON, NCT04538664) [94]. In addition, DZD9008, a selective, irreversible *EGFR*/*HER2* inhibitor, has been studied in two ongoing phase 1/2 studies (NCT03974022). In the phase 1 study, a favorable safety profile and ORR of 48.4% was observed [95]. Tarloxotinib, an agent developed as a prodrug that requires pathophysiologic hypoxic conditions for the activation of the irreversible dual *EGFR*/*HER2* TKI called tarloxotinib-E, has shown preclinical effectiveness in *EGFR* exon 20 insertions. However, the phase 2 study of tarloxotinib showed no response among 11 patients with an *EGFR* 20 insertion [96]. 

EGFR-TKI plus antibody therapy has been investigated, including cetuximab, necitumumab, and amivantamab, in combination with second- and third-generation EGFR-TKIs. The combination of afatinib and cetuximab has shown clinical efficacy in EGFR-mutant NSCLC; however, this combination has led to high rates of dermatologic toxicities [97]. Regarding exon 20 insertions, 3 out of 4 patients treated with afatinib plus cetuximab exhibited a partial response [19]. A phase 2 trial evaluating the efficacy of afatinib plus cetuximab for *EGFR* exon 20 insertions in NSCLC is currently ongoing (NCT03727724). In addition, a phase 1 study of necitumumab plus osimertinib (NCT02496663) and a phase 3 study involving amivantamab plus lazertinib are currently in progress (NCT04487080). Table 2 presents the clinical trial data of exon 20 insertion-targeted therapy.

## 6. Structural-Based Classification

EGFR-TKIs have been developed for EGFR-mutant NSCLC patients classified into exon-based groups. Recently, an alternative method for predicting the response to EGFR-TKIs was reported and was based on structural and functional changes [98]. A panel of 76 cell lines expressing *EGFR* mutations spanning exons 18–21 was created, and screening against 18 *EGFR* inhibitors representing first-, second-, and third-generation and exon 20 insertion-active EGFR-TKIs was conducted. Cell lines were stratified into four subgroups: classical-like mutations that were distant from the ATP-binding pocket, T790M-like mutations in the hydrophobic core, insertions in the loop at the C-terminal end of the αC-helix in exon 20, and mutations predicted to be P-loop and αC-helix compressing (PACC mutations). This approach better defines groups of mutations based on drug sensitivity, compared to the previously used exon-based classification. Classical-like mutations are sensitive for all generations of EGFR-TKIs, especially for third-generation TKIs. *EGFR* exon 20 insertions are known to present a heterogeneous response to EGFR-TKIs [87]. In the study, most exon 20 point-mutations were PACC mutations and were sensitive to second-generation EGFR-TKIs. Most exon 20 insertions in the αC-helix, on the other hand, behaved similarly to classical-like mutations and were pan-sensitive to EGFR-TKIs. The remaining mutations that occurred in the C-terminal loop of the αC-helix, known as Exon20ins-L mutations, were only sensitive to second-generation EGFR-TKIs. T790M-like mutants were classified into a third-generation TKI-sensitive (T790M-like-3S) subgroup and a third-generation TKI-resistant (T790M-like-3R) subgroup. T790M-like-3S mutants showed high selectivity for third-generation TKIs and some exon 20 insertion-active inhibitors, and moderate selectivity for anaplastic lymphoma kinase (ALK) and protein kinase C (PKC) inhibitors. T790M-like-3R mutants were resistant to classical EGFR-TKIs, however, they retained sensitivity to *ALK* and PKC inhibitors. PACC mutations showed significant selectivity for second-generation EGFR-TKIs compared to other TKIs. To determine whether structure-based groups could identify which class of TKI would be most beneficial to patients harboring atypical *EGFR* mutations, a retrospective analysis was conducted [98]. Patients harboring PACC mutations treated with second-generation EGFR-TKIs had a significantly longer TTF of 21.7 months than those treated with either first- or third-generation TKIs of 10.0 and 4.1 months, respectively (*p* < 0.0001, HR = 0.23). However, among patients harboring non-PACC mutations, the TTF was not significantly different between the classes of EGFR-TKIs.

## 7. Discussion

Clinical outcomes for patients harboring *EGFR* mutations in NSCLC have dramatically improved. Uncommon *EGFR* mutations are rare and highly heterogeneous, however, and there are no prospective clinical trials comparing the efficacy of different EGFR-TKIs or EGFR-TKI with chemotherapy as a result, except for a few trials such as PAPILLON and MARIPOSA trials for patients with exon 20 insertion. Therefore, there are no fixed first-line treatments recommended in the NCCN guidelines. According to the clinical results, however, afatinib and osimertinib would be treatment options as the first-line treatments for major uncommon mutations. The data from the prospective UNICORN study are limited to uncommon mutations, including compound mutations. For other minor uncommon mutations, physicians should consider available preclinical data, and real-world clinical data should include tolerability data to decide the treatment strategy.

In the exon 20 insertions, A763_Y764insFQEA is reported to be sensitive to first-, second-, and third-generation TKIs. Amivantamab, which was approved by the FDA in May 2021, showed encouraging data for patients with exon 20 insertions who had progressed on or after platinum-based chemotherapy. Considering its efficacy and AEs, amivantamab may be a good option for patients receiving second- or later-line therapies. The results of the ongoing MARIPOSA trial may have affected the first-line therapy. Mobocertinib was also approved by the FDA in September 2021 for patients who had progressed on or after platinum-based chemotherapy. However, since severe diarrhea was observed in 22% of patients, good management of AEs is required. A new classification to predict the sensitivity of EGFR-TKIs based on structural and functional changes provided new insights from the aspect of treatment development, especially for rare, uncommon mutations which make it difficult to obtain clinical data.

Combination therapies have been developed as first-line treatments for common mutations. Gefitinib plus platinum-doublet chemotherapy or erlotinib plus anti-angiogenic agents showed prolonged PFS. These may be treatment options for first-line treatments other than osimertinib. Although improvement of OS with combination therapy has not been observed in some studies, post-treatment may have a larger effect on the long OS due to treatment improvement for common mutation EGFR-TKIs. In many countries, osimertinib is the preferred first-line treatment. Cytotoxic chemotherapy is the recommended post-treatment after first-line osimertinib treatment. Since osimertinib, when used as a first- or second-line treatment, is known to lead to acquired resistance [99], a different treatment strategy may be necessary according to the treatment line and resistance mechanisms. For common mutation patients with T790M acquired resistance, first- and second-line EGFR-TKI followed by osimertinib has also shown prolonged OS [100].

## 8. Conclusions

The treatment strategy of NSCLC patients harboring *EGFR* mutations has developed dramatically following the discovery of first-generation EGFR-TKIs. Recently, many clinical trials have revealed the features of uncommon mutations and resistance mechanisms, which have contributed to drug development. For uncommon *EGFR* mutations excluding exon 20 insertion, considering treatment strategies based on structure-based classification may become a promising method, and amivantamab and mobocertinib will be a good treatment option regarding the efficacy for patients with exon 20 insertion. For common *EGFR* mutation, first-line osimertinib is one of the standards of care in many countries. However, for patients with T790M acquired resistance, first- and second-generation EGFR-TKI followed by osimertinib will be another treatment option to prolong the survival. Among patients with L858R mutation, erlotinib plus ramcirumab is a good treatment strategy compared to EGFR-TKI monotherapy from the RELAY trial. 

The exploration of the resistance mechanisms after treatment with EGFR-TKI therapy is necessary to determine subsequent treatments in the future. This indicates the importance of conducting re-biopsies of the tumor or liquid biopsies at the appropriate time. Targeted therapy, which has been investigated recently, or platinum-based chemotherapy would become an option according to the resistance mechanism, such as gene amplification/mutation of *MET*, *EGFR C797S*, and *HER3*, and histological transformation, after acquiring osimertinib resistance. For patients having no or unknown targeted resistance mutation, ABCP may be an option currently, and beyond, osimertinib strategy with platinum + pemetrexed for the patients having CNS metastasis at baseline (EPONA study) or regardless of CNS metastasis status (COMPEL study) for post-osimertinib treatment. Other than cytotoxic chemotherapy, clinical trials of second-generation EGFR-TKI followed by osimertinib, and of post-treatment including ICI, are also in progress. Recently, an allosteric inhibitor (JBJ-09-063), which binds to a different *EGFR* site than existing ATP-competitive EGFR-TKIs, has been developed for patients with acquired resistance [101]. Until now, we have considered treatment selection for EGFR-mutated NSCLC based on the alteration in the gene sequence (primary structure), however, soon, the direction of development for treatment will be based on the structure (tertiary structure). Understanding individual mutation features, including their structure, is needed to develop new agents and explore new treatment strategies.

## Figures and Tables

**Figure 1 cancers-14-02519-f001:**
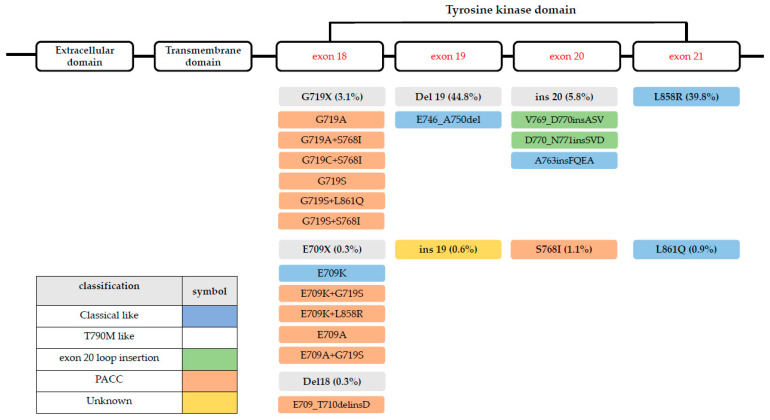
Structure of *EGFR* revealing common and uncommon mutations, compared with structural-based classification [18,19]. We have listed the mutations which are seen in more than 5% in each subgroup (*719X*, *E709X*, etc.), and assigned structural classifications.

**Table 1 cancers-14-02519-t001:** Frequency of uncommon *EGFR* mutations and drug sensitivity [52,54].

Exon	Mutation	Frequency (%)	In Vitro Sensitivity		
			Gefitinib/Erlotinib	Afatinib	Osimertinib
18	G719X	3.0–3.1	intermediate	sensitive	intermediate-sensitive
	E709X	0.3	E709K intermediate	sensitive	sensitive
	del 18	0.3	intermediate	sensitive	sensitive
19	del 19	44.8–45.0	sensitive	sensitive	sensitive
	ins 19	<0.6	intermediate	intermediate-sensitive	intermediate-sensitive
20	ins 20	>5.8	resistant except A763_Y764insFQEA	resistant-sensitive	intermediate-sensitive
	S768I	<1.5	intermediate	intermediate-sensitive	sensitive
21	L858R	35.0–39.8	sensitive	sensitive	sensitive
	L861Q	0.9–3.0	intermediate	intermediate-sensitive	intermediate-sensitive

Abbreviations: del, deletion; ins, insertion.

**Table 2 cancers-14-02519-t002:** Clinical trials of exon 20 insertion-targeted therapy.

Agents	Trial	Phase	Number of Patients	ORR (%)	PFS (Months)	OS (Months)	References
Mobocertinib	EXCLAIM	1/2	96	23	7.3 ^a^	24.0	[85,86]
Mobocertinib	EXCLAIM-2	3	NA	NA	NA	NA	NCT04129502
Poziotinib	ZENITH20-1	2	115	15	4.2	NA	[88]
TAS6417	NCT04036682	1/2	17	35	NA	NA	[89]
BDTX-189	NCT04209465	1/2	NA	NA	NA	NA	NCT04209465
Amivantamab	CHRYSALIS	1	81	40	8.3	NA	[92]
Amivantamab + chemotherapy	PAPILLON	3	NA	NA	NA	NA	NCT04538664
DZD9008	NCT03974022	1	97	48.4	NA	NA	[95]
Tarloxotinib	NCT03805841	2	11	0	NA	NA	[96]
Afatinib + cetuximab	NCT03727724	2	NA	NA	NA	NA	[19]
necitumumab + osimertinib	NCT02496663	1	NA	NA	NA	NA	NCT02496663
Amivantamab + lazertinib	NCT04487080	3	NA	NA	NA	NA	NCT04487080

Abbreviations: NA, not available; ORR, overall response rate; OS, overall survival; PFS, progression-free survival.

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
