# Peer review of "Treatment Strategies for Non-Small Cell Lung Cancer Harboring Common and Uncommon EGFR Mutations: Drug Sensitivity Based on Exon Classification, and Structure-Function Analysis"

_cancers, 2022, doi:10.3390/cancers14102519_

Round 1
Reviewer 1 Report
This article focused on the recent update of treatment strategies for NSCLC harboring EGFR mutation. In recent years, there are rapid development of next generation EGFR-TKI, EGFR-MET bispecific antibody, and emerging immunotherapy with new mechanism. Since various clinical studies using next generation EGFR-TKI, EGFR-MET bispecific antibody, antibody drug conjugate and emerging immunotherapy with new mechanism are being publish, this study addressed the question on what is the best treatment strategy at this moment.
This study is very informative and constructive. Especially, it included treatment strategy for resistance mechanism after first-line EGFR TKI. This study has the originality in terms of dealing with the most recent trends of NSCLC harboring EGFR mutation.
There is something disappointing about the conclusion part that it expressed too general (line 526-534). It would be better if the conclusion of the paper describes more about what the best strategy is in more detail.
It included the key article of a recent study. However, it had 97 references. This study focused on recent update, so it is better that remove old references.
Author Response
RESPONSES TO THE REVIEWERS
We appreciate reviewers’ insights. The reviewers’ comments helped us improve our manuscript, which we have revised according to the questions posed and recommendations offered by the reviewers.
Responses to Reviewer #1
There is something disappointing about the conclusion part that it expressed too general (line 526-534). It would be better if the conclusion of the paper describes more about what the best strategy is in more detail.
It included the key article of a recent study. However, it had 97 references. This study focused on recent update, so it is better that remove old references.
Response: Thank you for your valuable comments. We have modified the conclusion including the details of the treatment strategy regarded as a better option (line 532-534, and 540-559). Old references were mainly cited to understand the genomic backgrounds or pharmacology, and we have decided to remain them.
Reviewer 2 Report
The manuscript with title, “Treatment strategies for non-small cell lung cancer harboring EGFR mutations: Drug sensitivity based on structure-function analysis” has been reviewed. This paper is well written, and provides important topics regarding treatment strategies for non-small call lung cancer patients with both common and uncommon EGFR mutations. However, it does lack some important topics that are of interest to oncologists and cancer researchers, including:
- For patients with actionable EGFR mutations, any clinical outcome differences among treatment regimens including sequential EGFR-TKIs, concurrent EGFR-TKIs combined with platinum-doublet therapy, and sequential EGFR-TKIs followed by platinum-doublet therapy.
- For patients who acquired resistance to EGFR-TKIs, which strategies would be better choices, chemotherapy alone, chemotherapy plus anti-VEGF antibodies, or chemotherapy plus anti-VEGF antibodies in combination with immunotherapy. Although IMpower 150 gave some implications on this question, but it remained unclear whether chemotherapy combined with IO is better than chemotherapy alone. Are there any other clinical features which will potentially predict outcome, such as liver metastases?
Author Response
RESPONSES TO THE REVIEWERS
We appreciate reviewers’ insights. The reviewers’ comments helped us improve our manuscript, which we have revised according to the questions posed and recommendations offered by the reviewers.
Responses to Reviewer #2
Comment 1: For patients with actionable EGFR mutations, any clinical outcome differences among treatment regimens including sequential EGFR-TKIs, concurrent EGFR-TKIs combined with platinum-doublet therapy, and sequential EGFR-TKIs followed by platinum-doublet therapy.
Response: Thank you for your valuable comments. As you mentioned, the clinical outcome may differ among treatments, and several studies are ongoing. Osimertinib followed by second-generation EGFR-TKI (NEJ025B and NCT03757102) and osimertinib with chemotherapy (FLAURA2 study) is ongoing. JCOG1404, a study comparing gefitinib or osimertinib with inserted cisplatin and pemetrexed with gefitinib or osimertinib treatment as a first-line treatment is in progress. In addition, chemotherapy with or without osimertinib in patients who progressed after first-line Osimertinib is currently ongoing.
Comment 2: For patients who acquired resistance to EGFR-TKIs, which strategies would be better choices, chemotherapy alone, chemotherapy plus anti-VEGF antibodies, or chemotherapy plus anti-VEGF antibodies in combination with immunotherapy. Although IMpower 150 gave some implications on this question, but it remained unclear whether chemotherapy combined with IO is better than chemotherapy alone. Are there any other clinical features which will potentially predict outcome, such as liver metastases?
Response: Thank you for your valuable comments. We were not able to discuss the topic you have mentioned since this review focused on structure based treatment strategy. IMpower 150 trial showed prolonged PFS and OS in patients who received ABCP compared to those who received BCP and ACP. However, it was a subgroup analysis, and only ten pre-treated patients with EGFR mutation were included. ABCP may be the option when choosing IO regimen. As you mentioned, the efficacy of chemotherapy combined with IO is still unclear, and the results of other trials including EGFR- TKI pre-treated patients (NCT03256136) are awaited. We did not find any specific clinical features that predict the outcome. We have added the sentences referring to this topic in the conclusion (line 540-559).
Reviewer 3 Report
Kitadai et al reviewed treatment strategies for NSCLC with EGFR mutations. Overall, this is a timely topic but the authors tried to review everything from TKI to chemo to subtypes of mutations, and therefore lack focus and depth.
In addition, there was figure for the review article. The table 1 was just adapted from one article. No retrospective trials for uncommon EGFR mutation NSCLC were summarized.
Author Response
RESPONSES TO THE REVIEWERS
We appreciate reviewers’ insights. The reviewers’ comments helped us improve our manuscript, which we have revised according to the questions posed and recommendations offered by the reviewers.
Responses to Reviewer #3
Comment : Kitadai et al reviewed treatment strategies for NSCLC with EGFR mutations. Overall, this is a timely topic but the authors tried to review everything from TKI to chemo to subtypes of mutations, and therefore lack focus and depth. In addition, there was figure for the review article. The table 1 was just adapted from one article. No retrospective trials for uncommon EGFR mutation NSCLC were summarized.
Response: Thank you for your valuable comments. We have modified table 1 including another reference. We have excluded retrospective studies with a small sample size. Although we have mentioned some retrospective trials for uncommon EGFR mutation in the main text (references 50, 56), we did not cite trials regarding first-line EGFR-TKI, and we have added referring to it (line 301-303).
Reviewer 4 Report
Dear Authors,
Really nice review manuscript that earns to be published in high IF Journal! Correct, understandable structure, description, discussion and References part is simply amazing! Thank you very much, excellent jib!
There is only one small remark from my side regarding the methodological part of manuscript. Please, look at Lines 71-73 and ADD: key words for the literature search, data bases, time period, clear inclusion/exclusion criteria for the searched literature. This is that what seemed absent for me and I think that it is easily correctable!
Otherwise this is very good manuscript and I will advice to publish it as soon as this small thing would be corrected!
Author Response
RESPONSES TO THE REVIEWERS
We appreciate reviewers’ insights. The reviewers’ comments helped us improve our manuscript, which we have revised according to the questions posed and recommendations offered by the reviewers.
Responses to Reviewer #4
Comment : There is only one small remark from my side regarding the methodological part of manuscript. Please, look at Lines 71-73 and ADD: key words for the literature search, data bases, time period, clear inclusion/exclusion criteria for the searched literature. This is that what seemed absent for me and I think that it is easily correctable!
Response: Thank you for your comments. This review is not a systematic review or meta-analysis which is required for a comprehensive literature search. Also, our review is covered from basic investigation to clinical trials, therefore, it cannot adapt any criteria for literature search. However, we are really grateful your reviewing and constructing comment for our review.
Reviewer 5 Report
- Perhaps the title should be modified .... lung cancer harboring common and uncommon EGFR mutations.
- Does one paper regarding structure-function analysis warrant inclusion in the title
Author Response
RESPONSES TO THE REVIEWERS
We appreciate reviewers’ insights. The reviewers’ comments helped us improve our manuscript, which we have revised according to the questions posed and recommendations offered by the reviewers.
Responses to Reviewer #5
Comment 1: Perhaps the title should be modified …. lung cancer harboring common and uncommon EGFR mutations.
Response: Thank you for your comment. We have modified the title to “Treatment strategies for non-small cell lung cancer harboring common and uncommon EGFR mutations: Drug sensitivity based on exon classification, and structure-function analysis.”
Comment 2: Does one paper regarding structure-function analysis warrant inclusion in the title.
Response: Thank you for your comment. Since we have discussed both exon-based classification and structure-function analysis, we have modified the title to “Treatment strategies for non-small cell lung cancer harboring common and uncommon EGFR mutations: Drug sensitivity based on exon classification, and structure-function analysis.”
Round 2
Reviewer 3 Report
No real improvement.